# Increased Serum Mir-150-3p Expression Is Associated with Radiological Lung Injury Improvement in Patients with COVID-19

**DOI:** 10.3390/v14071363

**Published:** 2022-06-23

**Authors:** Larissa C. M. Bueno, Layde R. Paim, Eduarda O. Z. Minin, Luís Miguel da Silva, Paulo R. Mendes, Tatiana A. Kiyota, Angelica Z. Schreiber, Bruna Bombassaro, Eli Mansour, Maria Luiza Moretti, Jonathan Tak-Sum Chow, Leonardo Salmena, Otavio R. Coelho-Filho, Licio A. Velloso, Wilson Nadruz, Roberto Schreiber

**Affiliations:** 1Department of Internal Medicine, School of Medical Sciences, University of Campinas, Campinas 13083-887, SP, Brazil; larissacmbueno@gmail.com (L.C.M.B.); layde_rosane@yahoo.com.br (L.R.P.); dudaminin@gmail.com (E.O.Z.M.); ftluis_miguel@hotmail.com (L.M.d.S.); pramendes29@gmail.com (P.R.M.); talkiy74@gmail.com (T.A.K.); eliemansour27@hotmail.com (E.M.); lmoretti@unicamp.br (M.L.M.); tavicocoelho@gmail.com (O.R.C.-F.); lavellos@unicamp.br (L.A.V.); wilnj@unicamp.br (W.N.); 2Department of Clinical Pathology, School of Medical Sciences, University of Campinas, Campinas 13083-887, SP, Brazil; zaninele@gmail.com; 3Obesity and Comorbidities Research Center, University of Campinas, Campinas 13083-864, SP, Brazil; brunabombassaro@gmail.com; 4Department of Pharmacology and Toxicology, University of Toronto, Toronto, ON M5S 1A8, Canada; jonathants.chow@utoronto.ca (J.T.-S.C.); leonardo.salmena@utoronto.ca (L.S.)

**Keywords:** coronavirus, microRNA, SARS-CoV-2, lung injury

## Abstract

Coronavirus disease 2019 (COVID-19) is caused by the SARS-CoV-2 virus, responsible for an atypical pneumonia that can progress to acute lung injury. MicroRNAs are small non-coding RNAs that control specific genes and pathways. This study evaluated the association between circulating miRNAs and lung injury associated with COVID-19. **Methods:** We evaluated lung injury by computed tomography at hospital admission and discharge and the serum expression of 754 miRNAs using the TaqMan OpenArray after hospital discharge in 27 patients with COVID-19. In addition, miR-150-3p was validated by qRT-PCR on serum samples collected at admission and after hospital discharge. **Results:** OpenArray analysis revealed that seven miRNAs were differentially expressed between groups of patients without radiological lung improvement compared to those with lung improvement at hospital discharge, with three miRNAs being upregulated (miR-548c-3p, miR-212-3p, and miR-548a-3p) and four downregulated (miR-191-5p, miR-151a-3p, miR-92a-3p, and miR-150-3p). Bioinformatics analysis revealed that five of these miRNAs had binding sites in the SARS-CoV-2 genome. Validation of miR-150-3p by qRT-PCR confirmed the OpenArray results. **Conclusions:** The present study shows the potential association between the serum expression of seven miRNAs and lung injury in patients with COVID-19. Furthermore, increased expression of miR-150 was associated with pulmonary improvement at hospital discharge.

## 1. Introduction

Coronavirus 2019 disease (COVID-19) is a pandemic caused by the SARS-CoV-2 virus [1], responsible for an atypical pneumonia that may progress to acute lung injury, acute respiratory distress syndrome (ARDS), and pulmonary fibrosis [2]. The clinical features of COVID-19 are highly variable, ranging from asymptomatic patients to severe forms of respiratory failure. In this regard, computed tomography has been used as the main tool to diagnose COVID-19 pneumonia and its complications, enabling an accurate assessment of lung injuries [3].

MicroRNAs (miRNAs) are a class of small non-coding RNAs (22 nucleotides) with the function of controlling specific genes by modulating the pathways to which the target genes belong. miRNAs can modulate the function of various cells and are involved in the pathophysiology of responses to viral infections [4,5] and respiratory diseases [6,7]. In addition, these molecules have been used as biomarkers of diverse diseases [8,9].

The observation that the expression of various circulating miRNAs may be altered during COVID-19, as well as increasing evidence pointing to the role of miRNAs in SARS-CoV-2 infection [10], suggests that altered expression levels of circulating miRNAs may also be useful as biomarkers in the diagnosis and prevention of the disease. However, to date, little is known about the role of circulating miRNAs in the development of lung injury associated with COVID-19. This study is aimed at evaluating the association between circulating miRNAs and radiological pulmonary injury in patients with COVID-19.

## 2. Materials and Methods

### 2.1. Patient Recruitment, Characteristics, and Sample Acquisition

This study was carried out 27 patients with COVID-19, who were hospitalized at the Clinics Hospital of the State University of Campinas from April/2020 to June/2020 as part of a previously published study [11]. Inclusion criteria were age over 18 years, diagnosis of SARS-CoV-2 by the RT-PCR method according to the Berlin–Charité protocol [12], duration of symptoms of 12 days or less upon recruitment, diagnosis of COVID-19 typical pneumonia confirmed by chest computed tomography (CT), and SpO2 ≤ 94% in ambient air or Pa02/FiO2 ≤ 300 mmHg. The study followed the principles of the Declaration of Helsinki and was approved by the Research Ethics Committee of the State University of Campinas (CAEE protocol: 32076720.9.0000.5404). Written consent to participate in this study was obtained from all patients.

Clinical data were obtained at hospital admission, while CT scans were performed at admission and at discharge. Pulmonary CT scoring (Aquilion, Canon) was performed by two experienced radiologists according to guidelines published by Pan et al. [13]. The five lung lobes were classified at admission and at hospital discharge, with scores ranging from 0 to 5 according to the degree of involvement of the lung parenchyma with the following parameters: 0, no involvement; 1, <5% involvement; 2, 5–25% involvement; 3, 26–49% involvement; 4, 50–75% involvement; 5, >75% involvement. The scores of the five lung lobes were then summed, and the participants with global lung scores ranging from of 0–5, 6–10, 11–15, 16–20, and 21–25 were considered to have <5%, 5–25%, 26–49%, 50–75%, and >75% global lung involvement respectively. Patients were then categorized according to the degree of global lung involvement in: 1—Patients who showed radiological improvement in lung involvement at hospital discharge compared to admission; 2—Patient who did not show radiological improvement in lung parenchyma involvement.

Clinical characteristics were obtained at admission and 30 days after hospital admission (discharge) and comprised the following data: age, sex, blood pressure, smoking, hypertension, body mass index, diabetes, oxygen saturation, PAO_2_/FiO_2_ ratio, respiratory rate, duration of symptoms onset before admission, hospital unit where the patient was admitted, and whether the patient used supplementary oxygen or mechanical ventilation during the in-hospital stay. Blood glucose, C-reactive protein, high-sensitivity troponin I, brain natriuretic peptide, D-dimer, urea, and creatinine were assayed by standard techniques on 12-h fasting blood samples. Hematological parameters were determined by automated methods.

### 2.2. Extraction and Analysis of Serum miRNA Expression

The main objective of this report was to identify miRNAs that would be associated with COVID-19-induced lung injury. To achieve this aim, we collected serum samples of the patients 30 days after hospital admission to evaluate the expression of circulating miRNAs and to associate these measures with markers of lung injury. The miRNA samples were extracted using the miRNeasy Serum/Plasma kit (Qiagen, Valencia, CA, USA). The purity and quantity of isolated RNA were determined by OD260/280 using a NanoDrop ND-1000 Spectrophotometer (Thermo Fisher Scientific, Waltham, MA, USA). The expression of miRNAs was performed using the TaqMan OpenArray Human MicroRNA system (Thermo Fisher Scientific), which is capable of analyzing 754 microRNAs on a microfluidic platform in two sets of primer pools, panel A and B (Applied Biosystems, Waltham, MA, USA). Data analysis was performed using the Expression Suite Software version 1.0.4. (Thermo Fisher Scientific), and data were excluded from analyses with amplification scorse < 1.2 and cycle threshold (Ct) values > 35. Data were normalized using the manufacturer’s suggested global normalization method and previous reports [14].

Serum expression of miR-150-3p was measured by qRT-PCR in samples collected at admission and 30 days after hospital admission. Reverse transcription (RT) reactions were performed on a Mastercycler ep (Eppendorf, Hamburg, Germany) 96-Well Thermal Cycler according to the manufacturer’s instructions and performed using the SuperScript^®^III First Strand Synthesis Kit (Applied Biosystems, Waltham, MA, USA). Reactions were performed in triplicate on StepOne Plus Real-Time PCR Systems (Thermo Fisher Scientific, Inc.). The reactions were heated at 95 °C for 10 min, followed by 40 cycles of 95 °C for 15 s and 60 °C for 1 min. Data were normalized using U6 snRNA (noncoding small nuclear RNA-001973) as the housekeeping gene. The comparative Ct (ΔΔCt) method to quantify relative gene expression was used, and fold change (FC) was calculated as FC = 2 − ΔΔCt, where Ct is defined as the PCR cycle number at which the fluorescence meets the threshold in the amplification plot [15].

### 2.3. Target Prediction in SARS-CoV-2 Genome and Receptors

We used RNA22 as a miRNA prediction tool to identify the potential miRNA response element (MRE) in the SARS-CoV-2 viral genome and potential binding sites of miRNAs on ACE2 and TMRSS2 receptors. This computational tool can predict the heteroduplex formed by the miRNA and the target RNA sequence [16], and in this way we can identify the miRNA binding region in the virus genome and in the viral receptors in the human cell.

### 2.4. Statistical Analysis

Clinical and laboratory characteristics of patients are reported as mean ± standard deviation (SD) and median (25th,75th percentiles) for continuous variables with normal or non-normal distribution and frequencies and percentages for categorical data. For continuous variables with normal or non-normal distribution, the Student’s *t* test and Mann–Whitney test were used, respectively. The chi-square test was used to assess the frequency differences between the groups. Differential expression of miRNAs profile by OpenArray was considered significant when there was a relative fold change >1.5 or <–1.5 and a *p* value ≤ 0.05. To detect differences in miR-150-3p expression by qRT-PCR, serum miRNA expression was compared between two groups (with radiological pulmonary improvement and without radiological pulmonary improvement) using the Mann–Whitney test, while for multiple comparisons, we used the Kruskall–Wallis test followed by the Wilcoxon test. A *p*-value < 0.05 was considered statistically significant. SPSS 15.0 software was used for statistical analyses.

## 3. Results

### 3.1. Clinical Characteristics of Participants and miRNA Expression Levels

Patients (mean age = 51 years, 44% women) were included in our study from April 2020 to 27 June 2020. At hospital discharge, 16 patients had radiological pulmonary improvement, while 11 patients presented no radiological pulmonary improvement (Appendix A). The median (25th, 75th percentiles) time between the CT scans was 11 days [8,14]. There were no differences in the characteristics of the patients at admission, except for a greater global lung CT scoring in the group with radiological lung improvement (Appendix A). Table 1 presents the characteristics of the participants at hospital discharge according to pulmonary involvement. Except for greater blood lymphocyte count and lower serum creatinine levels in the groups with pulmonary improvement, there were no clinical or laboratory differences at discharge between the two groups.

To identify miRNAs associated with lung injury caused by COVID-19, we initially analyzed a panel of 754 miRNAs from serum samples collected 30 days after hospital admission and identified the expression of 211 miRNAs after quality control screening and 43 miRNAs expressed in more than 50% of the samples (Figure 1). OpenArray analysis (Figure 2A,B) revealed that seven miRNAs were differentially expressed between groups of patients without radiological pulmonary improvement compared to those with radiological pulmonary improvement after hospital discharge, with three miRNAs being significantly upregulated (miR-548c-3p, miR-212-3p, and miR-548a-3p) and four significantly downregulated (miR-191-5p, miR-151a-3p, miR-92a-3p, and miR-150-3p).

### 3.2. Target Prediction in SARS-CoV-2 Receptors

In order to identify between the two main SARS-CoV-2 receptors in the human cell, namely the angiotensin converting enzyme 2 (ACE2) and the transmembrane serine protease 2 (TMPRSS2) potential binding sites with the differentially expressed miRNAs in our study, an in silico analysis using the RNA22 target prediction tool was performed, and only miR-191-5p resulted in a potential binding site with the ACE2 receptor, while miR-150-3p resulted in three potential binding sites with TMPRSS2 (Appendix A).

### 3.3. Target Prediction in SARS-CoV-2

Using the RNA22 target prediction tool, we sought to identify, among the seven miRNAs with differential expression by OpenArray, potential binding sites to target the SARS-CoV-2 reference genome (NC_045512.2, Figure 3). This analysis resulted in the identification of potential binding sites for five of these miRNAs. We did not identify binding sites for miR-548a and miR-548c in the SARS-CoV-2 genome. miRNA-150-3p had seven binding sites identified in the viral genome, while miR-212-3p and miR-151a-3p had two binding sites each. miRNA-191-5p and miR-92a-3p each had one binding site in the viral genome. The position of the binding sites of miRNAs predicted by the RNA22 tool are shown in Appendix A.

### 3.4. Validation of miR-150-3p Expression Levels

Literature reports on the role of miR-150 in viral infection [17,18,19] led us to a better assessment of miR-150-3p expression by qRT-PCR in COVID-19 patients, comparing their serum levels at admission and 30 days after hospital admission. Validation by qRT-PCR confirmed the results obtained by OpenArray (Figure 4A) in the sample collected 30 days after admission. Furthermore, when we compared the serum expression of miR-150-3p by qRT-PCR in samples from patients with COVID-19 at admission and 30 days after hospital admission, we observed an increase in the serum expression of this miRNA in patients with improved lung disease at hospital discharge, while patients without radiological pulmonary improvement showed no statistically significant difference when compared to hospital admission (Figure 4B).

### 3.5. Correlation Analysis

The correlation analysis between miRNAs differentially expressed by OpenArray (30 days after hospital discharge) and laboratory parameters in the same period showed an inverse correlation between lymphocyte counts and miR-191-5p, miR-548a-3p and miR- 548c-3p, and miR-191-5p with serum creatinine, in addition to a direct correlation between miR-548a-3p with serum creatinine and miR-548c-3p with D-Dimer levels (Appendix A). However, these correlations lost significance after adjusting for potential confounders such as gender, age, BMI, medication, length of hospital stay, hypertension, and diabetes.

The only miRNA evaluated in samples obtained at admission by qRT-PCR, i.e., miR-150-3p, showed an inverse correlation with lymphocyte counts and a direct correlation with platelet levels (Appendix A), that remained significant after adjustment for possible variables of confusion such as gender, age, BMI, medication, length of stay, hypertension, and diabetes (Beta ± SE= −3.877 ± 1.529, *p* = 0.023; 46.345 ± 19.258, *p* = 0.028 respectively).

## 4. Discussion

In the present study, we evaluated the serum expression of 754 miRNAs in 27 patients with COVID-19, and their association with SARS-CoV-2-induced lung injury after hospital discharge. Of these, seven miRNAs, including miR-150-3p, were differentially expressed in patients without radiological pulmonary improvement when compared to those with radiological pulmonary improvement.

We did not find in the literature studies relating miR-150-3p, miR-191-5p, miR-151a-3p, miR-92a-3p, miR-548c-3p or miR-548a-3p, differentially expressed in the present article, with ACE2 receptor or transmembrane serine protease 2 (TMPRSS2). However, three studies [20,21,22] showed that miR-212-5p is related to the ACE2 receptor, with increased expression in patients with COVID-19, mainly in the most severe cases [20]. In the present study, in silico analysis indicated that the ACE2 receptor has a potential binding site with miR-191-5p, while TMPRSS2 showed thee potential binding sites with miR-150-3p.

A recent report in the literature points to a reduction in the expression of miR-150-5p in patients with COVID-19 when compared to individuals without infection and to an interaction of this miRNA with miRNA recognition sites (MRE) in the nsp10 gene, which is crucial for replication and evasion of the host’s immune response [17]. Furthermore, Xu et al. [18] demonstrated a reduction in inflammation and pulmonary edema with increased expression of miR-150 in mice with non-COVID-19-lung injury. Meanwhile, Chow and Salmena [19] identified 128 human miRNAs with the potential to target the SARS-CoV-2 genome, including miR-150-3p and miR-151a-3p. Our in silico analysis demonstrated the presence of at least seven binding sites of miR-150-3p in the SARS-CoV-2 genome, in genes such as M.N, nsp4, nsp13, nsp15, ORF3a, and ORF7, implying a large interference potential of this miRNA in viral infection. In this regard, we validated this miRNA by qRT-PCR in patients with COVID-19, comparing the serum levels at hospital admission with those collected 30 days after hospital discharge, and we observed an increase in the serum expression of miR-150 in patients with radiological pulmonary improvement, while no significant changes in the expression of this miRNA was detected among patients without radiological pulmonary improvement. Therefore, based on our data, it is possible to speculate that pulmonary worsening after viral infection is associated with a reduction in miR-150 expression.

Two recent studies [23,24] suggested an increase in miR-92a expression in SARS-CoV-2 infection. Kazan et al. [23] reported the increased plasma expression of eight miRNAs, including miR-92a, in patients with COVID-19 when compared to healthy individuals, while Park et al. [24] demonstrated in cell culture that this increase would be related to a possible antiviral response. In our study, the expression of miR-92a was lower among patients without radiological pulmonary improvement, suggesting that an inefficient antiviral response of the organism was associated with reduced levels of this miRNA. Furthermore, our in silico analysis suggested the presence of a binding site of this miRNA in the genome of the SARS-CoV-2 virus, in the nsp10 gene. However, more studies are necessary to confirm the role of miR-92a in lung injury caused by SARS-CoV-2.

Increased expression of miR-212-5p was reported to be associated with greater severity of severe acute respiratory syndrome before the advent of COVID-19 [25], but to date no studies have demonstrated the involvement of miR-212-3p, as well as the miR-191-5p, miR-151a-3p, miR-548a, and miR-548c associated with COVID-19. However, our in silico analysis predicted the presence of at least two binding sites for miR-212-3p, in the nsp13 and nsp14 genes, two binding sites for miR-151a-3p, in the S and N genes, and a site of binding of miR-191-5p to the nsp13 gene of the SARS-CoV-2 virus, suggesting that these miRNAs may have some impact on the modulation of viral infection. We did not detect binding sites for miR-548a or miR-548c in the virus genome, and further studies should be performed to assess the impact of these miRNAs on COVID-19.

We acknowledge the limitations of our study. First, the sample size was relatively small. Second, our screening analysis of miRNs expression was based on samples collected 30 days after hospital admission, which were not simultaneously compared to the second chest CT scan, which was performed at hospital discharge. Therefore, it is possible that the expression of serum miRNAs levels shown herein may not necessarily reflect their levels at hospital discharge. Third, we cannot rule out the possibility that other miRNAs are related to worsening lung function in patients with COVID-19, because only seven of the 43 miRNAs present in more than 50% of the samples showed differential expression, despite 211 miRNAs being expressed in at least one of the samples. Fourth, several participants in the “without improvements” group started with 5–25 involvement, which may represent an early diagnosis. However, we found that the period between symptoms onset and hospital admission time was similar between the studied groups, making this hypothesis less probable. Fifth, the time between CT scans showed a marked range among the studied patients, which might have influenced our findings. On the other hand, this study is the first, as far as we know, to evaluate the expression profile of circulating miRNAs, with the potential to be used as a biomarkers of lung injury in patients with COVID-19.

## 5. Conclusions

The present study showed the potential association between the serum expression levels of seven miRNAs in patients with COVID-19 and pulmonary impairment. In addition, it demonstrated that patients who had increased expression of miR-150 during the course of SARS-CoV-2 infection had better pulmonary evolution. Further studies including a larger number of patients should be performed to confirm our findings and valuate the exact role of miR-150 in SARS-CoV-2-induced lung injury.

## Figures and Tables

**Figure 1 viruses-14-01363-f001:**
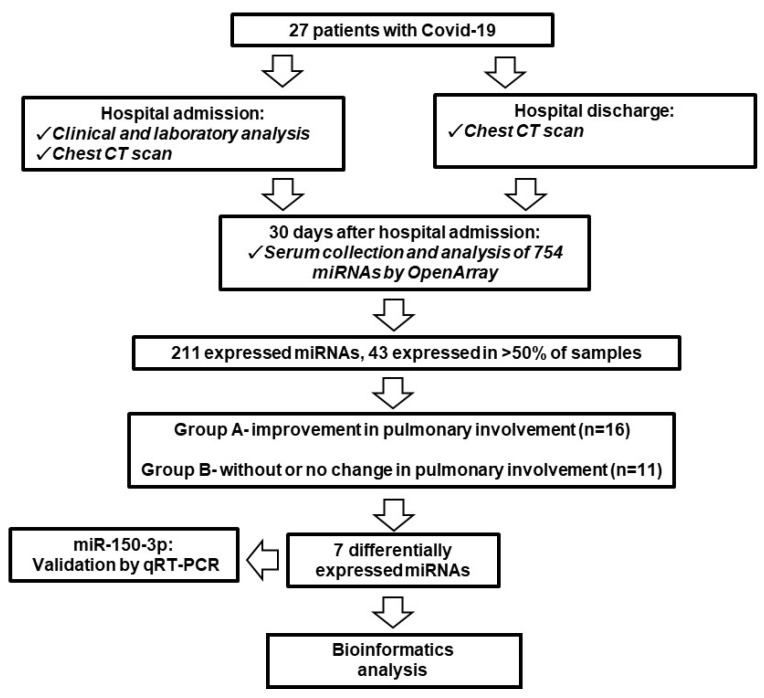
Flowchart illustrating the steps in the study of serum miRNA expression in pulmonary impairment in patients with COVID-19.

**Figure 2 viruses-14-01363-f002:**
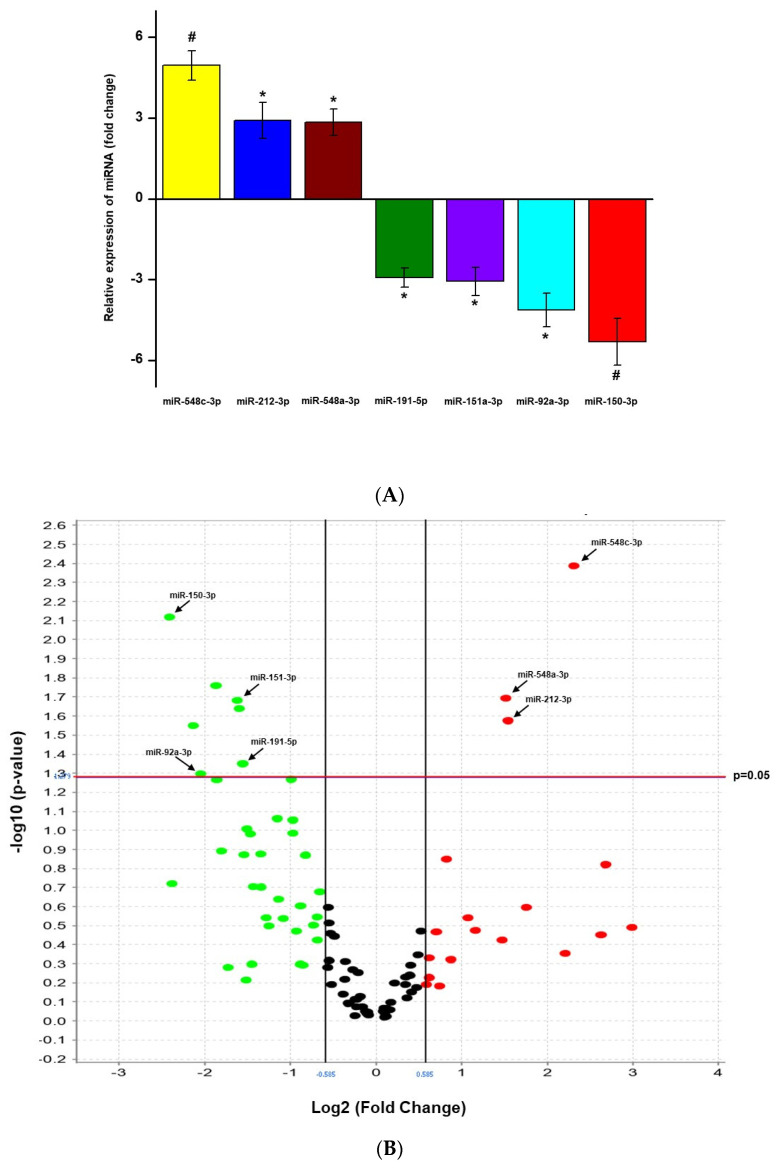
(**A**). Expression of microRNAs (miRNAs) in COVID-19 patients without radiological pulmonary improvement after hospital discharge, compared to patients with radiological pulmonary improvement (reference). Data are expressed as Fold change. Positive values mean upregulation and negative values mean downregulation. * *p* < 0.05 and # *p* < 0.01. (**B**). Differential expression analysis of microRNAs (miRNAs) obtained in the serum of COVID-19 patients without radiological pulmonary improvement after hospital discharge, compared to patients with radiological pulmonary improvement (reference). Volcano plot depicts the significantly altered miRNAs found (*p* < 0.05). Each dot represents an miRNA: upregulated are in red (log2 fold change ≥ 1.5), downregulated in green (log2 fold-change ≤ 1.5), and unchanged in black.

**Figure 3 viruses-14-01363-f003:**
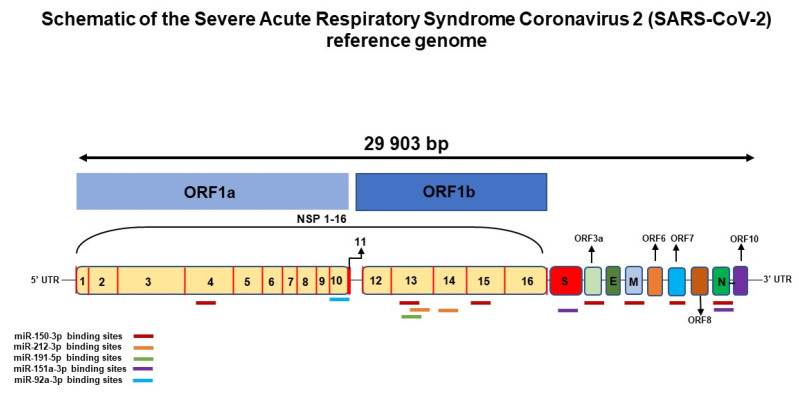
Schematic of the Severe Acute Respiratory Syndrome Coronavirus 2 (SARS-CoV-2) reference genome (NC_045512.2). Computational identification of predicted miRNA response elements (MREs) in the SARS-CoV-2 reference genome. Colored bars represent binding sites for miRNAs in the virus genome.

**Figure 4 viruses-14-01363-f004:**
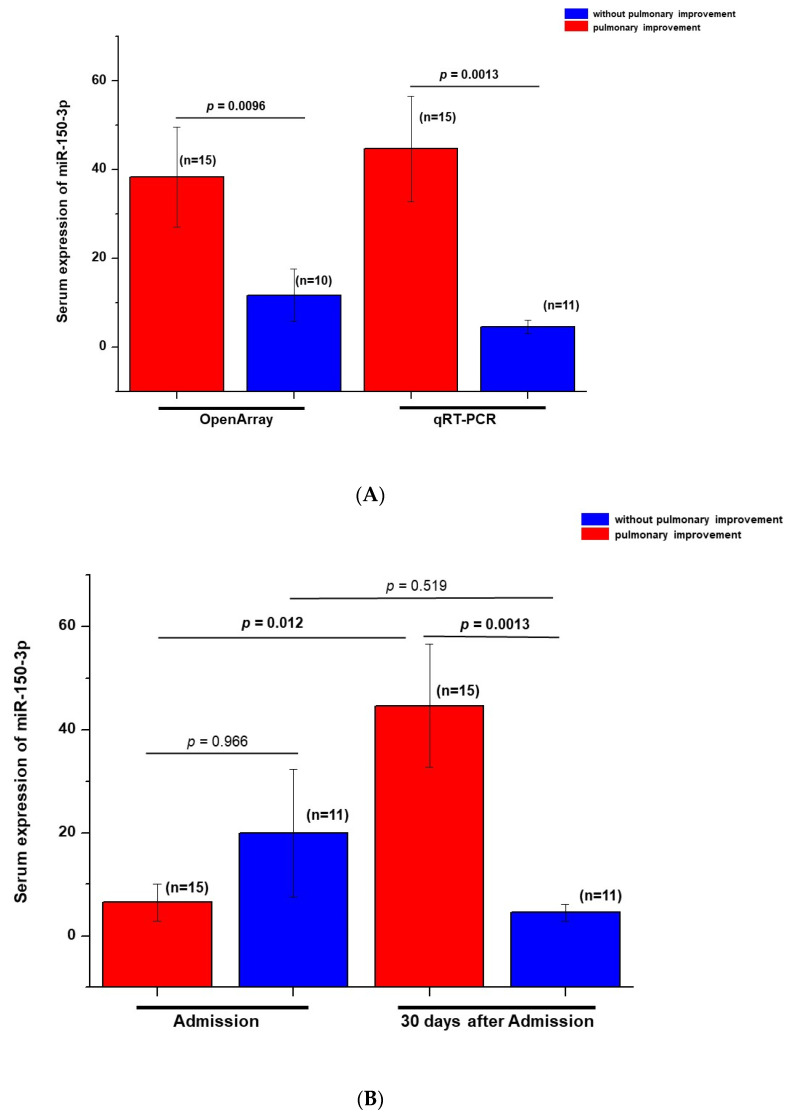
(**A**). Validation of OpenArray miR-150-3p results by qRT-PCR using serum samples obtained at 30 days after hospital admission. The Mann–Whitney two-tailed U test was used to compare groups. (**B**). Comparison by qRT-PCR of the serum expression of miR-150-3p in samples from patients with COVID-19 at admission and 30 days after hospital admission. The Kruskal–Wallis test followed by the Wilcoxon test was used to compare groups.

**Table 1 viruses-14-01363-t001:** Characteristics of participants at discharge according to radiological pulmonary improvement.

Parameter	With Improvement (*n* = 16)	Without Improvement (*n* = 11)	*p*
*Clinical characteristics*			
Female/male	9/7	3/8	0.273
Age (years)	49.1 ± 13.4	53.6 ± 8.0	0.322
Body Mass Index (kg/m^2^)	32.5 ± 4.6	30.0 ± 4.5	0.187
Hypertension, *n* (%)	6 (37.5)	5 (45.5)	0.988
Diabetes mellitus, *n* (%)	6 (37.5)	4 (36.4)	0.952
Obesity, *n* (%)	12 (75.0)	7 (63.6)	0.836
Former smoker, *n* (%)	0 (0)	2 (18.2)	0.355
Systolic blood pressure (mmHg)	139.0 ± 17.6	134.7 ± 16.7	0.532
Diastolic blood pressure (mmHg)	67.9 ± 34.9	76.2 ± 34.5	0.550
Oxygen saturation (%)	97.8 ± 1.1	97.6 ± 0.9	0.669
PaO2/FiO2 ratio	314 [272–335]	335 [310–371]	0.095
Respiratory rate, per minute	20.8 ± 3.8	19.0 ± 2.4	0.176
*Laboratory data*			
White cell count (×10^9^/L)	7.1 [6.2–8.8]	6.0 [5.5–6.6]	0.054
Lymphocyte count (×10^9^/L)	4.8 [1.9–30.7]	1.60 [1.2–5.1]	0.020
Platelet count (×10^9^/L)	334.3 ± 84.0	290.1 ± 143.5	0.322
Plasma glucose (mg/dL)	134.5 ± 47.8	128.7 ± 47.7	0.771
Serum creatinine (mg/dL)	0.64 [0.50–0.83]	0.86 [0.77–1.04]	0.009
AST (U/L)	30 [17–42]	28 [21–60]	0.477
C-reactive protein (mg/L)	15.8 [3.7–34.0]	23.1 [9.8–39.7]	0.278
High-sensitivity troponin I, ng/L	8.5 [4.9–24.8]	7.6 [4.2–13.7]	0.712
Brain natriuretic peptide, ng/mL	50 [50–83]	50 [50–499]	0.765
D-dimer, μg/mL	1254 [643–2029]	780 [672–999]	0.256
Urea (mg/dL)	26 [21–31]	28 [26–47]	0.217
*Global lung CT scoring*			
Score	11.2 ± 4.3	12.1 ± 3.4	0.569

Abbreviations: AST, aspartate aminotransferase. Patient data were compared using Student’s *t* test and Mann–Whitney test for continuous variables with normal or non-normal distribution, respectively. The chi-square test was used to assess frequency differences between groups. *p* < 0.05 was considered statistically significant.

## Data Availability

All data presented in this study are available upon request for correspondence from the author, and the raw data will be archived in an Institutional Data Repository.

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
