# Peer review of "Increased Serum Mir-150-3p Expression Is Associated with Radiological Lung Injury Improvement in Patients with COVID-19"

_viruses, 2022, doi:10.3390/v14071363_

Round 1

Reviewer 1 Report

To the authors:

The manuscript titled:” INCREASED SERUM MIR-150-3P EXPRESSION IS ASSOCIATED WITH LUNG INJURY IMPROVEMENT IN PATIENTS WITH COVID-19” by Larissa C.M. Bueno evaluates the association between circulating miRNAs and lung injury due to COVID-19.

Major comments:

  • The major problem of the manuscript is the definition of “pulmonary improvement” based only on qualitative subjective analysis. The authors could also have used the percentage of lung normally aerated, poorly aerated and non-aerated. I also suggest to use functional data as: PaO2/FiO2 and spirometric data.
  • 9/11 patients in the “without improvements” group started with 5-25 involvement, which may represent an early diagnosis. Moreover the degree of lung involvement at discharge look quite similar between the groups. Moreover the Time between CT scans (days) shows a huge range. Please comment.
  • Which were the discharge indications? Where similar for all the patients?
  • Performing an analysis looking to the degree of involvement should be quite informative.
  • Baseline data are poor. I suggest adding: PaO2/FiO2, Respiratory rate, O2 therapy, ventilatory support and number of days from the start of symptoms.
  • Clinical data at discharge are missing
  • The category 0-5 and 0-25 of involvement of the lung parenchyma, overlap. They should not (see patient L10, L5)

Minor comments:

  • In which ward were the patients admitted?

Author Response

Replies to reviewers.

Reviewer 1

“The major problem of the manuscript is the definition of “pulmonary improvement” based only on qualitative subjective analysis. The authors could also have used the percentage of lung normally aerated, poorly aerated and non-aerated. I also suggest to use functional data as: PaO2/FiO2 and spirometric data.”

Answer: We thank the reviewer for the insightful and pertinent comments.

First, we apologize for the lack of clarification regarding the method used to assess COVID-induced lung involvement. Actually, CT scan scoring was performed by two experienced and skilled radiologists and followed previously published guidelines (1). In addition, this method was also cited in the original paper that described the sample that was studied in the current report (2). For CT scoring, the radiologists evaluated the presence of ground glass opacity, consolidation and crazy-paving pattern. Initially, each of the five lung lobes was scored 0–5 according to the following parameters: 0, no involvement; 1, <5% involvement; 2, 5–25% involvement; 3, 26–49% involvement; 4, 50–75% involvement; 5, >75% involvement. Then, the scores of the 5 lung lobes were summed and the participants with global lung scores ranging from of 0-5, 6-10, 11-15, 16-20, 21-25 were considered to have <5%, 5–25%, 26–49%, 50–75% and >75% global lung involvement, respectively. In the current paper, lung involvement categorization was estimated based on the global lung scores. Furthermore, in the new version of the manuscript, we also included the global lung scores in the studied groups at admission and discharge. Conversely, given that the major endpoint of our study was to evaluate the impact of COVID-19 infection on the lungs as assessed by CT, we clarified this issue by indicating that we categorized the patients according to "radiological pulmonary improvement" rather than "pulmonary improvement" as seen in line 78 (2.1. Patient Recruitment, Characteristics, and Sample Acquisition).

Second, we thank the reviewer for the suggestion of evaluating the percentage of lung normally aerated, poorly aerated and non-aerated. However, this technique uses a semi-automated method (Thoracic Volume ComputerAssisted Reading software, GE Health care, United States) (3), which we do not have access. Therefore, we did not perform this analysis. In addition, due to COVID-19 restrictions imposed by the hospital where the study was performed, spirometry was not performed in any patient admitted to the hospital during the early months of the pandemic. For this reason, we had no participant enrolled in the study who performed spirometry when treated for SarsCov-2 infection.

Third, in accordance with the reviewer's suggestions, we have included functional respiratory data in the new version of the manuscript. Please see data in Tables 1 and Supplement S2.

“9/11 patients in the “without improvements” group started with 5-25 involvement, which may represent an early diagnosis. Moreover the degree of lung involvement at discharge look quite similar between the groups. Moreover the Time between CT scans (days) shows a huge range. Please comment.”

Answer: We agree with the reviewer that several participants in the “without improvements” group started with 5-25 involvement, which may represent an early diagnosis. However, we found that the period between symptoms onset and hospital admission time was similar between the studied groups ("with improvement” vs “without improvement” = 8.5 [6-10] vs 8 [7-9] days; p=0.753), turning this hypothesis less probable. These novel data are presented in new Supplemental Table S2 and are commented in the Discussion Section as one of our limitations (line 305).

We agree with the reviewer that the degree of lung involvement was similar between the studied groups at discharge, even though there was a greater burden of lung alterations in the "with improvement” group in comparison with the “without improvement” group at admission, as shown in new data regarding global lung CT scores that are now presented in new Table 1 and Supplemental Table S2.

We also agree with the reviewer that the Time between CT scans (days) shows a huge range among the studied patients, which might have influenced our findings. This topic is addressed in new Discussion section as one of the limitations of our study (line 309).

“Which were the discharge indications? Where similar for all the patients?”

Answer: Patients were discharged when they were clinically stable, with improved symptoms and were not requiring oxygen supplementation. The discharge indications were similar for all patients.

"Performing an analysis looking to the degree of involvement should be quite informative."

Answer: We agree with the reviewer that the degree of lung involvement was similar between the studied groups at discharge, even though there was a greater burden of lung alterations in the "with improvement” group in comparison with the “without improvement” group at admission, as shown in new data regarding global lung CT scores that are now presented in new Table 1 and Supplemental Table S2.

“Baseline data are poor. I suggest adding: PaO2/FiO2, Respiratory rate, O2 therapy, ventilatory support and number of days from the start of symptoms.”

Answer: Thanks to the reviewer for pointing out this relevant issue. In the new version of the manuscript, these data were included in new Table 1 and Supplemental Table S2.

“Clinical data at discharge are missing”

Answer: Thanks to the reviewer for pointing out this issue. We agree with the reviewer that Table 1 was confusing and clinical admission data should be presented when in fact we presented data at hospital discharge. In the new version of the manuscript, Table 1 presents the characteristics of the patients at discharge, while novel supplementary table S2 presents the characteristics of the patients at admission. We appreciate the reviewer’s comments and apologize for our confusion.

“The category 0-5 and 0-25 of involvement of the lung parenchyma, overlap. They should not (see patient L10, L5)”

Answer: We apologize for any flaws in the revision of the former version of the manuscript. The correct values of involvement of the lung parenchyma are: <5; 5-25; 26-49; 50-75 and >75, as previously reported (2). In the new version of the manuscript and in the supplementary material, typing errors have been corrected (line 77).

“In which ward were the patients admitted?”

Answer: These data were added to new Table Supplemental S2 and showed no differences in the rate of admission at the ward or ICU between the two studied groups.

References

1- Pan, F.; Ye, T.; Sun, P.; Gui, S.; Liang, B.; Li, L.; Zheng, D.; Wang, J.; Hesketh, R.L.; Yang, L.; Zheng, C. Time Course of Lung Changes at Chest CT during Recovery from Coronavirus Disease 2019 (COVID-19). Radiology. 2020 Jun;295(3):715-721. doi: 10.1148/radiol.2020200370.

2- Mansour, E.; Palma, A.C.; Ulaf, R.G.; Ribeiro, L.C.; Bernardes, A.F.; Nunes, T.A.; Agrela, M.V.; Bombassaro, B.; Monfort-Pires, M.; Camargo, R.L.; Araujo, E.P.; Brunetti, N.S.; Farias, A.S.; Falcão, A.L.E.; Santos, T.M.; Trabasso, P.; Dertkigil, R.P.; Dertkigil, S.S.; Moretti, M.L.; Velloso, L.A. Safety and Outcomes Associated with the Pharmacological Inhibition of the Kinin-Kallikrein System in Severe COVID-19. Viruses 2021, 13, 309. doi.org/10.3390/v13020309.

3- Daisuke Yamada, Sachiko Ohde, Ryosuke Imai, Kengo Ikejima, Masaki Matsusako, Yasuyuki Kurihara. Visual classification of three computed tomography lung patterns to predict prognosis of COVID-19: a retrospective study. BMC Pulm Med. 2022 Jan 3;22(1):1)

Reviewer 2 Report

The authors identified miR-150 as an indicator of lung injury in COVID-19 infection. It is quite an interesting article; however, I believe these suggestions might improve it.

1- More information should be added in the introduction about the role of circulating miRNAs in COVID-19

2- miRNAs are altered and affected by the presence of comorbidities such as diabetes and hypertension. Did the authors try to investigate the effect of these factors on the investigated miRNA?

3- miRNA expression could be affected by the different treatments of COVID-19 patients. Is there a difference between the medications taken by the 2 investigated groups?

4- The authors normalize the expression of miRNAs to U6; however U6 is not suitable for the normalization of serum miRNAs. Authors need to reanalyze the data using an endogenous miRNA control or using the delta CT and not the delta delta CT method.

5- It would be crucial if the authors could perform a correlation analysis between miRNA expression at admission with the laboratory tests done at the same time. Further, if possible to be done at both time points.

6- In table 1, the numbers of brain natriuretic peptides should be either range or mean and SD. Also, table 1 legend should be more descriptive.

7- In Figures 4A and 4B, the comparison of miR-150-3p between the 2 groups showed different p values, once p=0.0013 and another p=0.005. Shouldn't it be the same?

8- The authors performed bioinformatics with the viral genome, however, it would be interesting if any of these miRNAs could also bind to host entry receptors such as ACE2 or TMPRSS. Authors could investigate that using in silico approaches.

9- Out of all the 7 miRNAs, the authors focused on miR-150. This needs to be justified. The panel of 7 miRNAs would provide a greater impact as a panel for lung injury in COVID-19.

10- miR-212 was known to be affected by severity as well as obesity, gender, and comorbidities such as diabetes. The authors could mention that in the discussion section.

Author Response

Replies to reviewers.

Reviewer 2

1- More information should be added in the introduction about the role of circulating miRNAs in COVID-19

Answer: We thank the reviewer for the insightful and pertinent comments and as suggested, we added a paragraph in the intro section on the role of circulating miRNAs in covid-19 (line 50).

2- miRNAs are altered and affected by the presence of comorbidities such as diabetes and hypertension. Did the authors try to investigate the effect of these factors on the investigated miRNA?

Answer: We agree with the reviewer that miRNAs can be affected by comorbidities such as diabetes and hypertension, as these comorbidities are risk factors for the severity of Covid-19. We performed correlation analysis between these clinical variables and miRNAs differentially expressed by OpenArray and miR-150-3p evaluated by qRT-PCR at admission and at discharge. We did not observe any correlation between them and, therefore, these data were not included in this article, but are presented below.

Bivariate correlation coefficients between miRNAs differentially expressed by OpenArray and comorbidities.

miRNAs

Diabetes mellitus

Hypertension

Obesity

Gender

r

p

r

p

r

p

r

p

Log miR-150-3p

-0.045

0.830

- 0.170

0.417

-0.012

0.955

-0.122

0.561

Log miR-92a-3p

0.098

0.625

- 0.232

0.244

-0.073

0.718

-0.057

0.776

Log miR-151a-3p

-0.068

0.783

- 0.154

0.528

0.207

0.394

0.029

0.905

Log miR-191-5p

0.179

0.381

0.053

0.798

-0.388

0.051

0.337

0.092

Log miR-548a-3p

-0.079

0.719

0.255

0.240

0.234

0.283

-0.066

0.764

Log miR-548c-3p

0.110

0.609

0.037

0.865

-0.051

0.813

-0.260

0.220

The correlation of log-transformed expression of miRNAs with comorbidities was assessed by Spearman's Method.

Bivariate correlation coefficients between miR-150-3p differentially expressed by qRT-PCR and comorbidities.

miRNAs

Diabetes mellitus

Hypertension

Obesity

Gender

r

p

r

p

r

p

r

p

Log miR-150-3p (admission)

-0.084

0.682

- 0.119

0.561

0.389

0.959

-0.288

0.154

Log miR-150-3p (discharge)

0.137

0.504

- 0.171

0.403

-0.133

0.516

0.031

0.881

3- miRNA expression could be affected by the different treatments of COVID-19 patients. Is there a difference between the medications taken by the 2 investigated groups?

Answer: We thank the reviewer for the opportunity to clarify this point. Patients were treated with routine medication in the period from April/June 2020. The only difference was related to the use of bradykinin inhibitors as published in Viruses (1). In our study, the use of these inhibitors did not affect the expression of differentially expressed miRNAs. In new supplementary table S1 we included the drugs used by each patient.

4- The authors normalize the expression of miRNAs to U6; however U6 is not suitable for the normalization of serum miRNAs. Authors need to reanalyze the data using an endogenous miRNA control or using the delta CT and not the delta delta CT method.

Answer: We thank the reviewer for the opportunity to clarify this point. The lack of a standard endogenous control for miRNA expression studies in any human tissue remains egregious. Ideally, suitable endogenous normalizers should be selected according to the experimental conditions, considering the type of disease, population and type of biological material (2). U6 snRNA is often used as a normalizer in miRNA profiling studies (3-5), considering its high stability due to its short hook structure and lack of (6) and its widespread use as load control in published studies (4) even in human serum samples. In a recent study, Huang et al (7) studied miR-155 expression in serum samples from patients with acute respiratory distress syndrome using U6 as an endogenous control. All samples analyzed in our study had U6 expression detected. Considering that this study, by using serum samples from patients with a clinical picture of respiratory failure similar to that of patients with Covid-19 and the results of the stability of U6 in our samples (see table below), we consider that this RNA can be used in our validation study of miR-150-3p as an endogenous control. We take the opportunity to emphasize that the analysis by OpenArray, which provided the profile of 7 differentially expressed miRNAs, uses the global normalization method (8) as directed by the manufacturer.

U6 admission

Ct mean

With Improvement

30.50

Without improvement

31.62

p

0.258

U6 30 days after hospital admission

With Improvement

33.83

Without improvement

33.87

p

0.540

5- It would be crucial if the authors could perform a correlation analysis between miRNA expression at admission with the laboratory tests done at the same time. Further, if possible to be done at both time points.

Answer: Thanks to the reviewer for the suggestion. We had not thought of doing this analysis and the suggestion certainly helped to improve the quality of our article. In samples analyzed by OpenArray (30 days after hospital discharge) and laboratory parameters in the same period, we observed an inverse correlation between lymphocytes and miR-191-5p, miR-548a-3p and miR-548c-3p. Furthermore, we observed a correlation between miR-191-5p (inverse) and miR-548a-3p (direct) with serum creatinine. We also verified a correlation between D-dimer and miR-548c-3p (direct). However, these correlations lost significance after adjusting for possible confounders, such as gender, age, BMI, medication, length of hospital stay, hypertension, and diabetes (supplemental table S5). The other laboratory parameters showed no correlation with the differentially expressed miRNAs evaluated by OpenArray. The only miRNA evaluated in samples obtained at admission by qRT-PCR, ie miR-150-3p, showed an inverse correlation with lymphocytes and a direct correlation with platelet levels that remained significant after adjustment for possible confounding variables, such as gender, age, BMI, medication, length of stay, hypertension and diabetes (supplemental table S6). These new results were incorporated into the manuscript in section: 3.5. Correlation analysis (line 231) and in supplementary material such as supplementary table S5 and S6.

6- In table 1, the numbers of brain natriuretic peptides should be either range or mean and SD. Also, table 1 legend should be more descriptive.

Answer: Thanks to the reviewer for pointing out this error. In the new version this error was fixed. We also fixed the table 1 legend.

7- In Figures 4A and 4B, the comparison of miR-150-3p between the 2 groups showed different p values, once p=0.0013 and another p=0.005. Shouldn't it be the same?

Answer: Thanks to the reviewer for pointing out this error. The correct value is 0.0013. In the new version the typo has been corrected. We apologize for this error.

8- The authors performed bioinformatics with the viral genome, however, it would be interesting if any of these miRNAs could also bind to host entry receptors such as ACE2 or TMPRSS. Authors could investigate that using in silico approaches.

Answer: Thanks to the reviewer for the suggestion. As suggested by the reviewer, an in silico analysis was performed and only miR-191-5p resulted in a potential binding site with ACE2 receptor, while miR-150-3p resulted in 3 potential binding sites with serine protease 2 transmembrane (TMPRSS2). We did not find in the literature studies relating miR-150-3p; miR-191-5p; miR-151a-3p; miR-92a-3p; miR-548c-3p or miR-548a-3p differentially expressed in the present article with ACE2 receptor or transmembrane serine protease 2 (TMPRSS2). however, 3 studies (9-11) showed that miR-212-5p is related to the ACE2 receptor with increased expression in patients with covid-19, mainly in most severe cases (9). These results were incorporated into our manuscript in the methods (line 118), results (line 231), discussion (line 252) and in the supplementary material (table S3).

9- Out of all the 7 miRNAs, the authors focused on miR-150. This needs to be justified. The panel of 7 miRNAs would provide a greater impact as a panel for lung injury in COVID-19.

Answer: We thank the reviewer for the opportunity to clarify this point. The present study is a hypothesis launch and presented a profile of 7 differentially expressed miRNAs analyzed by the OpenArray system that can evaluate 754 miRNAs. The decision to validate miR-150-3p was due to the fact that two independent studies drew the attention to the differential expression of miR-150-5p in patients with COVID-19 (12) and its relationship with inflammation and pulmonary edema in alternative models of lung injury (13). Furthermore, bioinformatics analysis demonstrated that miR-150-3p has 7 potential binding sites in the SARSCoV-2 genome (14). Our study suggests that further investigations are needed to be carried out with a greater number of patients in order to validate the role of these miRNAs in pulmonary evolution.

10- miR-212 was known to be affected by severity as well as obesity, gender, and comorbidities such as diabetes. The authors could mention that in the discussion section.

Answer: We did not observe a correlation between miR-212 and gender or comorbidities such as obesity and diabetes, and therefore these data were not included in this article but are presented below.

Correlation analysis between miR-212 expression by OpenArray with the laboratory tests 30 days after hospital discharge.

miRNAs

Diabetes mellitus

Hypertension

Obesity

Gender

r

p

r

p

r

p

r

p

Log miR-212-3p

-0.075

0.774

-0.120

0.646

0.026

0.920

-0.241

0.352

References

  1. Mansour, E.; Palma, A.C.; Ulaf, R.G.; Ribeiro, L.C.; Bernardes, A.F.; Nunes, T.A.; Agrela, M.V.; Bombassaro, B.; Monfort-Pires, M.; Camargo, R.L.; Araujo, E.P.; Brunetti, N.S.; Farias, A.S.; Falcão, A.L.E.; Santos, T.M.; Trabasso, P.; Dertkigil, R.P.; Dertkigil, S.S.; Moretti, M.L.; Velloso, L.A. Safety and Outcomes Associated with the Pharmacological Inhibition of the Kinin-Kallikrein System in Severe COVID-19. Viruses 2021, 13, 309. doi.org/10.3390/v13020309.

  1. Muhammad Yogi Pratama, Luisa Cavalletto, Claudio Tiribelli, Liliana Chemello, Devis Pascut. Selection and validation of miR-1280 as a suitable endogenous normalizer for qRT-PCR Analysis of serum microRNA expression in Hepatocellular Carcinoma. Sci Rep. 2020 Feb 21;10(1):3128. doi: 10.1038/s41598-020-59682-0.

  1. Lu Fang, Andris H Ellims, Xiao-lei Moore, David A White, Andrew J Taylor, Jaye Chin-Dusting, Anthony M Dart. Circulating microRNAs as biomarkers for diffuse myocardial fibrosis in patients with hypertrophic cardiomyopathy. J Transl Med. 2015 Sep 24;13:314. doi: 10.1186/s12967-015-0672-0.

  1. M'Hammed Aguennouz, Fabrizio Guarneri, Rosaria Oteri, Francesca Polito, Roberta Giuffrida, Serafinella P Cannavò. Serum levels of miRNA-21-5p in vitiligo patients and effects of miRNA-21-5p on SOX5, beta-catenin, CDK2 and MITF protein expression in normal human melanocytes. J Dermatol Sci. 2021 Jan;101(1):22-29. doi: 10.1016/j.jdermsci.2020.10.014.

  1. Zhifeng Ma, Ting Zhu, Haiyong Wang, Bin Wang, Linhai Fu, Guangmao Yu. Investigation of serum markers of esophageal squamous cell carcinoma based on machine learning methods. J Biochem. 2022 Apr 13;mvac030. doi: 10.1093/jb/mvac030.

  1. G Shumyatsky, D Wright, R Reddy. Methylphosphate cap structure increases the stability of 7SK, B2 and U6 small RNAs in Xenopus oocytes. Nucleic Acids Res. 1993 Oct 11;21(20):4756-61. doi: 10.1093/nar/21.20.4756.

  1. Zhenfei Huang, Hui Huang, Meirong Shen, Changrong Li, Chao Liu, Huayong Zhu, Weiwei Zhang. MicroRNA-155-5p modulates the progression of acute respiratory distress syndrome by targeting interleukin receptors. Bioengineered. 2022 May;13(5):11732-11741. doi: 10.1080/21655979.2022.2071020.

  1. Pieter Mestdagh, Pieter Van Vlierberghe, An De Weer, Daniel Muth, Frank Westermann, Frank Speleman, Jo Vandesompele. A novel and universal method for microRNA RT-qPCR data normalization. Genome Biol. 2009;10(6):R64. doi: 10.1186/gb-2009-10-6-r64.

  1. Noha Mousaad Elemam, Hind Hasswan, Hayat Aljaibeji, Narjes Saheb Sharif-Askari, Rabih Halwani, Jalal Taneera, Nabil Sulaiman. Profiling Levels of Serum microRNAs and Soluble ACE2 in COVID-19 Patients. Life (Basel). 2022 Apr 12;12(4):575. doi: 10.3390/life12040575.

  1. Noha Mousaad Elemam, Hind Hasswan, Hayat Aljaibeji, Nabil Sulaiman. Circulating Soluble ACE2 and Upstream microRNA Expressions in Serum of Type 2 Diabetes Mellitus Patients. Int J Mol Sci. 2021 May 17;22(10):5263. doi: 10.3390/ijms22105263.

  1. Abeedha Tu-Allah Khan, Zumama Khalid, Hafsa Zahid, Muhammad Abrar Yousaf, Abdul Rauf Shakoori. A computational and bioinformatic analysis of ACE2: an elucidation of its dual role in COVID-19 pathology and finding its associated partners as potential therapeutic targets. J Biomol Struct Dyn. 2022 Mar;40(4):1813-1829. doi: 10.1080/07391102.2020.1833760.

  1. Shaw M Akula, Paul Bolin, Paul P Cook. Cellular miR-150-5p may have a crucial role to play in the biology of SARS-CoV-2 infection by regulating nsp10 gene. RNA Biol. 2022;19(1):1-11. doi: 10.1080/15476286.2021.2010959.

  1. Jiaxin Xu, Dan Xu, Zhizhong Yu, Zhaohui Fu, Zheng Lv, Lei Meng, Xin Zhao. Exosomal miR-150 partially attenuated acute lung injury by mediating microvascular endothelial cells and MAPK pathway. Biosci Rep. 2022 Jan 28;42(1):BSR20203363. doi: 10.1042/BSR20203363.

  1. Jonathan Tak-Sum Chow, Leonardo Salmena. Prediction and Analysis of SARS-CoV-2-Targeting MicroRNA in Human Lung Epithelium. Genes (Basel). 2020 Aug 26;11(9):1002. doi: 10.3390/genes11091002.
